# Association of *HOXC8* Genetic Polymorphisms with Multi-Vertebral Number and Carcass Weight in Dezhou Donkey

**DOI:** 10.3390/genes13112175

**Published:** 2022-11-21

**Authors:** Xiaoyuan Shi, Yan Li, Tianqi Wang, Wei Ren, Bingjian Huang, Xinrui Wang, Ziwen Liu, Huili Liang, Xiyan Kou, Yinghui Chen, Yonghui Wang, Faheem Akhtar, Changfa Wang

**Affiliations:** Agricultural Science and Engineering School, Liaocheng University, Liaocheng 252059, China

**Keywords:** *HOXC8*, Dezhou donkey, genetic polymorphism, carcass weight, vertebrae number

## Abstract

An increase in the number of vertebrae can significantly affect the meat production performance of livestock, thus increasing carcass weight, which is of great importance for livestock production. The homeobox gene C8 (*HOXC8*) has been identified as an essential candidate gene for regulating vertebral development. However, it has not been researched on the Dezhou donkey. This study aimed to verify the Dezhou donkey *HOXC8* gene’s polymorphisms and assess their effects on multiple vertebral numbers and carcass weight. In this study, the entire *HOXC8* gene of the Dezhou donkey was sequenced, SNPs at the whole gene level were identified, and typing was accomplished utilizing a targeted sequencing genotype detection technique (GBTS). Then, a general linear model was used to perform an association study of *HOXC8* gene polymorphism loci, multiple vertebral numbers, and carcass weight for screening candidate markers that can be used for molecular breeding of Dezhou donkeys. These findings revealed that *HOXC8* included 12 SNPs, all unique mutant loci. The *HOXC8* g.15179224C>T was significantly negatively associated with carcass weight (CW) and lumbar vertebrae length (LL) (*p* < 0.05). The g.15179674G>A locus was shown to be significantly positively associated with the number of lumbar vertebrae (LN) (*p* < 0.05). The phylogenetic tree constructed for the Dezhou donkey *HOXC8* gene and seven other species revealed that the *HOXC8* gene was highly conserved during animal evolution but differed markedly among distantly related animals. The results suggest that *HOXC8* is a vital gene affecting multiple vertebral numbers and carcass weight in Dezhou donkeys, and the two loci g.15179224C>T and g.15179674G>A may be potential genetic markers for screening and breeding of new strains of high-quality and high-yielding Dezhou donkeys.

## 1. Introduction

In recent years, people’s awareness of health care has improved, and the market for healthy food such as donkey meat, donkey skin, donkey bone, and donkey milk has expanded, leading to increasing demand for donkey products. Meanwhile, since donkeys experience long intergenerational intervals, most of them are single-birth breeding, which leads to the current situation of demand exceeding supply. It was essential to use molecular marker-assisted breeding to meet the market demand and speed up the selection process of Dezhou donkeys. The Dezhou donkey is one of China’s five premium donkey breeds with solid adaptability, rough feeding resistance, and disease resistance. It is an important livestock and poultry genetic resource and the primary source of donkey products [1,2]. We have assembled a high-quality reference genome for Dezhou donkeys [3], which provided a reference basis for exploring the effects of gene polymorphisms on phenotypic traits and the screening of molecularly assisted breeding markers.

The multi-vertebral number was mainly concentrated in the thoracic and lumbar vertebrae [4]. It was common in livestock populations and significantly impacted meat production [5,6]. Carcass weight is considered one of the most valuable economic traits in livestock production [7,8,9]. To date, studies investigating the effects of multi-vertebral number and carcass weight at the genetic level have focused on animals such as pigs [10,11], cattle [12,13], and sheep [6,14], while fewer studies have been conducted on Dezhou donkeys. Studies on pigs, yaks, and sheep have demonstrated that multi-vertebral numbers contribute to livestock body length [12,15,16]. The body length of livestock was positively correlated with carcass weight and hide production [17]. Based on our previous studies, it has been confirmed that gene single nucleotide polymorphisms do cause changes in body size, vertebrae number, and production performance in Dezhou donkeys. For example, the nuclear receptor subfamily 6 group A member 1 (*NR6A1*) g.18114954C>T can significantly increase the number of lumbar vertebrae, the total number of thoracolumbar vertebrae, and carcass weight in Dezhou donkeys, which can be a candidate gene to explore the genetic mutation affecting vertebral development [18]. However, the research is not sufficient and more genes need to be examined to screen for more molecular markers to accelerate the selection process of new strains of Dezhou donkeys.

Homeobox gene C8 (*HOXC8*), a member of the homeobox genes (*HOX*) family, is a highly conserved evolutionary transcription factor commonly found in vertebrates. It has been demonstrated that the *HOX* gene was a major regulatory gene in the development of vertebrate and mammalian body axis skeletons and was highly conserved during animal evolution [19,20]. Gene mutation experiments have confirmed that mutations in the mouse *HOXC8* gene induce the addition of a thoracic vertebra and an extra pair of ribs [21]. Differential expression of the *HOXC8* gene similarly caused vertebral changes in livestock, such as the appearance of multi-vertebral variants in pigs and Mongolian sheep [22,23]. Furthermore, *HOXC8* also played a regulatory role in the muscle tissue of adult mice and adipose differentiation in sheep [24,25]. Therefore, it is hypothesized that the *HOXC8* gene may be associated with the multi-vertebral number and carcass weight traits of Dezhou donkeys.

In keeping with the above views, this study aimed to detect mutations in the *HOXC8* gene, investigate the correlation of its genetic variation with multi-vertebral number and carcass weight traits, and provide potential candidate markers for the selection and breeding of new strains of high-quality and high-yielding Dezhou donkey.

## 2. Materials and Methods

### 2.1. Moral Statement

The Animal Ethics Committee approved all experimental animals used in this study of Liaocheng University (LC2019-1), and all conformed to animal welfare requirements.

### 2.2. Animals and DNA Samples

Blood samples of 400 adult male Dezhou donkeys with phenotypic data were collected from the jugular vein by venipuncture in a donkey farm (Yucheng, Shandong, China). Also, all of them were 2 years old fattening donkeys. Phenotypic records included body height (BH), body length (BL), chest circumference (CC), number of thoracic vertebrae (TN), number of lumbar vertebrae (LN), total number of thoracic and lumbar vertebrae (TLN), thoracic vertebrae length (TL), lumbar vertebrae length (LL), carcass weight (CW) and hide weight (HW). The body size measurements were performed with the donkeys standing on a level ground and in a natural stance, while all measurements were performed by the same operator to reduce human errors. The specific measurement method was described in detail in the article by Zhang et al. [26]. Genomic DNA was extracted using the Blood Genomic DNA Extraction Kit, dissolved in Elution Buffer TB, and stored at 4 degrees.

### 2.3. Identification of HOXC8 Gene Polymorphism

The mass determination of genomic concentrations was performed using an ultra-micro spectrophotometer (B-500, METASH, Shanghai, China). Then, all 400 eligible genomic samples (n > 1000 ng) were sent to Shijiazhuang MOL BREEDING Biotechnology Co., Ltd. (Shijiazhuang, China) for high-depth resequencing of the *HOXC8* gene using targeted sequencing genotype testing technology (GBTS) to examine genetic locus variants in the whole gene range precisely. Six samples were randomly selected to verify the accuracy of GBTS sequencing results, which were matched to three mutation sites by PCR amplification. Then, the PCR products were sent to Sangon Biotech (Shanghai) Co., Ltd. (Shanghai, China) for Sanger sequencing. Finally, the gene’s single nucleotide polymorphisms (SNPs) were identified in Dezhou donkeys and subsequently searched in the Ensembl database (https://asia.ensembl.org/Equus_asinus_asinus/Gene/Summary?db=core;g=ENSEASG00005010151;r=PSZQ01003681.1:11208231-11210397;t=ENSEAST00005015832, accessed on 24 May 2022), European Variation Archive (https://www.ebi.ac.uk/eva/?Home, accessed on 9 November 2022) and NCBI database (https://www.ncbi.nlm.nih.gov/search/all/?term=donkey, accessed on 9 November 2022) to check if these loci were novel mutations.

### 2.4. Validation of Polymorphic Loci

For better results, three pairs of primers were designed using Primer Premier 5.0 software (version 5.0, PREMIER Biosoft International, San Francisco, CA, USA) based on the *HOXC8* gene (GenBank NC_052198.1). More information on the primers is shown in Table 1. And then, six DNA samples were randomly selected as templates for PCR amplification. The total volume of PCR was 25 μL, including 12.5 μL 2× Taq PCR Master Mix (TIANGEN, without-dye, MF002, Beijing, China), 8.5 μL ddH_2_O, 1 μL each of upstream and downstream (Sangon Biotech, Shanghai, China), and 2 μL of 50 ng/μL DNA template. The PCR conditions were as follows: pre-denaturation at 95 °C for 10 min; followed by 30 cycles of denaturation at 95 °C for 30 s, annealing for 30 s (primer annealing temperature as shown in Table 1), extension at 72 °C for 30 s; and then extension at 72 °C for 10 min to complete the reaction. After the reaction, the PCR products were subjected to 2% agarose electrophoresis in 1× TBE (Tris-borate-EDTA) buffer and visualized by ethidium bromide staining. The PCR products were sent to Sangon Biotech Co., Ltd. (Shanghai, China) for sanger sequencing.

### 2.5. Statistical Analysis

Allele and genotype frequencies of mutant loci in Dezhou donkeys were calculated using SHEsis online software (http://analysis.bio-x.cn/myAnalysis.php, accessed on 24 May 2022) [27]. We also used the GDIcall Online Calculator (http://www.msrcall.com/Gdicall, accessed on 24 May 2022) to estimate population genetic parameters, including adequate allele number (*Ne*), population heterozygosity (*H*), observed heterozygosity (*Ho*), expected heterozygosity (*He*) [28] and the polymorphic information content (*PIC*) [29]. The distribution of genotypes at each locus was tested for compliance with Hardy-Weinberg equilibrium (HWE) using a chi-square test. Associative analysis of phenotypic traits with SNPs was performed using the general linear model of SAS version 9.4 (SAS Institute Inc., Cary, NC, USA). The statistical model is shown as follows:Y = µ + G + TL + XG + e

Y is the observed phenotypic values, µ is the population mean, G is the fixed effect of genotype, TL is the fixed effect of types of thoracolumbar vertebrae, XG is the genotype-related interactions and was removed from the models when found to be not significant (*p* > 0.05), and e is each observation’s random error effect. Since the experimental population was consistent in age, sex, season of birth, feeding environment and season of sample collection, their effects were not considered in the model. SAS version 9.4 software (SAS Institute Inc., Cary, NC, USA) was also used to correlate phenotypic traits with types of thoracolumbar. The analysis model is: Y = μ + e, where Y represents individual phenotypic records, the μ represents the population mean, and e represents the random error. All phenotypic data were tested for normal distribution using the Shapiro-Wilk test. Phenotypic data that did not conform to a normal distribution were transformed with Box-Cox or Johnson.

### 2.6. Phylogenetic Analysis

A phylogenetic tree of *HOXC8* gene sequences in eight species (Dezhou donkey, pig, horse, mouse, cattle, goat, human, and zebrafish) was constructed by the maximum likelihood method with MEGA-X software 11.0 [30]. For Dezhou donkey, pig, horse, mouse, cattle, goat, human, and zebrafish, sequences were obtained from the NCBI (https://www.ncbi.nlm.nih.gov/, accessed on 25 May 2022) Reference Sequences (NC_052198.1, NC_010447.5, NC_009149.3, NC_000081.7, NC_037332.1, NC_030812.1, NC_052198.1, NC_000012.12, and NC_007134.7, respectively).

## 3. Results

### 3.1. Accuracy of GBTS

The sequencing profiles of the three mutant loci are shown in Figure 1. The results indicated that the sequencing results of the three mutant loci matched the sequencing results of GBTS, and the genotypes were consistent with the liquid-phase microarray typing results.

### 3.2. Analysis of HOXC8 Gene Polymorphisms and Population Genetics in Dezhou Donkey

By mapping to the known Dezhou donkey gene sequence (GenBank NC_052198.1), 12 mutant loci were identified in the *HOXC8* gene, and all loci were novel. The mutation information for the *HOXC8* gene is summarized in Table 2. The alignment analysis of *HOXC8* revealed 10 SNPs downstream and 2 in the introns. Among the 12 mutant loci, non-diallelic loci and those with genotypic typing failure were filtered out, leaving 4 mutant loci for subsequent analysis (g.15177692G>A, g.15178184C>T, g.15179224C>T, g.15179674G>A). Population genetic analysis was performed on four SNPs of the *HOXC8* gene. The results revealed that all four loci (g.15177692G>A, g.15178184C>T, g.15179224C>T, g.15179674G>A) had low polymorphism (*PIC* < 0.25) in Dezhou donkeys (Table 3). In addition, the chi-square test indicated that the other three loci of *HOXC8* were in Hardy-Weinberg equilibrium (*p* < 0.05), except for the g.15179674G>A locus (*p* > 0.05).

### 3.3. Association Analysis

Association analysis between the SNPs and phenotypes revealed that the g.15179224C>T locus was significantly associated with CW (*p* < 0.01). It was also significantly associated with LL (*p* < 0.05). The g.15179674G>A locus was significantly associated with LN (*p* < 0.05). More information is shown in Table 4. The g.15179224C>T locus was not observed as a mutant homozygote. At this locus, the CW of wild homozygotes of Dezhou donkeys was significantly higher than that of mutant heterozygotes *(p* < 0.01). Its LL was considerably higher than mutant heterozygous genotype individuals (*p* < 0.05). It indicates that mutation at this locus may be detrimental to the increase of CW and LL. There were three genotypes of the g.15179674G>A locus. The LN was significantly higher in individuals with mutant homozygotes than in wild homozygotes and mutant heterozygotes (*p* < 0.05). It indicates that mutations at this locus may be conducive to increased LN. Analysis of the association between thoracolumbar types and phenotypic traits showed that thoracolumbar types were significantly correlated with carcass weight. These individuals with T17L5 showed minimal values. The detailed information is shown in Table 5.

### 3.4. Construction of Phylogenetic Tree of HOXC8 Gene for Dezhou Donkey and Other Seven Species of Animals

To investigate the variation of the *HOXC8* gene in different species, a phylogenetic tree of the *HOXC8* gene was constructed by the neighbor-joining method, and 1000 bootstrap replications were performed, showing that donkeys and horses were most closely related (Figure 2). The donkey, horse, cattle, goat, pig, and human sequences were clustered together and differed significantly from those of other breeds. The results showed that the *HOXC8* gene was highly conserved in mammals. The phylogenetic tree indicated that the *HOXC8* gene was highly conserved during animal evolution but differed considerably among distantly related species.

## 4. Discussion

Numerous studies have shown that the *HOXC8* gene was essential for skeletal morphogenesis, hematopoiesis, and cartilage differentiation during embryonic development [31,32]. Meanwhile, *HOXC8* was also involved in regulating the development and progression of various human malignancies, such as gastric and breast cancer [33,34,35]. In addition, the *HOXC8* gene has been associated with vertebral development in mice [36], myogenesis in Korean Hanwoo cattle [37], kyphosis and arthrogryposis multiplex congenital (AMC) in pigs [38,39], and the multi-vertebral number trait in Mongolian sheep [40]. *HOXC8* was also widely expressed in the muscle, reproductive, and digestive systems in adult Dezhou donkeys [41]. However, no studies have been conducted on the *HOXC8* polymorphism in donkeys. Therefore, we explored the association between the polymorphism of the *HOXC8* gene and multi-vertebral number and carcass weight traits in donkeys.

In this study, we undertook an association study of four mutant loci in 400 individual Dezhou donkeys and identified two SNPs associated with CW, LL, and LN traits at the whole gene level (*p* < 0.05). Here, we presented four SNPs associated with each attribute (Table 4). The g.15179224C>T and g.15179674G>A were located in the introns of the *HOXC8* gene. Only the genotype CC and CT were present at g.15179224C>T. Individuals with the CT genotype had significantly lower CW and LL than those with the CC genotype (*p* < 0.05), and BL, CC, HW, LN, and TL had the same downward trend but without reaching significant levels. Therefore, this locus was an adverse mutation for CW and LL. Consequently, it was speculated that the function of this locus would probably be to reduce BL by affecting LL and then to reduce CW by reducing BL. All three genotypes were present at g.15179674G>A, GG, GA, and AA, respectively. Individuals with the AA genotype had significantly higher LN than those with GG and GA genotypes (*p* < 0.05). In contrast, BH, CC, and HW had the opposite trend, though none reached a significant level. Therefore, the mutation at this locus was a favorable mutation for LN, which was unfavorable for BH, CC, and HW. Since the mutation sites affecting CW, LL and LN in Dezhou donkeys were all located on introns, their specific regulatory mechanisms were unclear. Studies have shown that mutations on introns cause alterations in splice variants [42,43], so *HOXC8* gene expression might be regulated by the variable shear mechanism. The g.15179224C>T and g.15179674G>A sites may alter the binding of shear factor binding proteins to target sequences, leading to the appearance of phenotypic variants. Furthermore, the association of the *HOXC8* gene with CW in Dezhou donkeys might be related to adipose differentiation and lipid deposition during livestock development and the regulation of LL and LN by the *HOXC8* gene [18,44,45]. The correlation of the *HOXC8* gene with LL and LN was possibly attributed to the differential expression of the *HOXC8* gene during skeletal differentiation.

Analysis of the phylogenetic tree showed that the Dezhou donkey was most closely related to horses, with the sequences of donkey, horse, cattle, goat, pig, and humans all clustered together. According to the findings, the *HOXC8* gene was highly conserved throughout animal evolution, which was in line with the previous conclusion of Krumlauf [46]. However, zebrafish did not cluster with them, probably due to the significant genetic sequence differences caused by the distant affinities.

Though the *HOXC8* gene polymorphism of the Dezhou donkey was studied for the first time, some limitations to our study should be highlighted. Several studies have confirmed that *HOXC8* was not a single gene that influenced the developmental process of the vertebra but most likely in combination with other genes or regulatory elements [47]. In contrast, we only investigated the effect of *HOXC8* single gene polymorphism on multiple vertebral numbers in Dezhou donkeys, and the study of genetic mechanisms was not comprehensive enough. In addition, the results may have been underestimated due to sample size constraints. Therefore, more mutant loci, gene interactions, and experimental validation with larger sample sizes should be evaluated to understand better the effects of genetic polymorphisms on multi-vertebral numbers and carcass weight traits. In the meantime, it is worth exploring whether the increase in the number or length of thoracolumbar vertebrae may have some unintended negative consequences. Based on current studies in domestic animals, no direct clinical signs of increased number or length of thoracolumbar vertebrae have been reported, and specific welfare issues need to be explored.

## 5. Conclusions

In this study, we identified 12 mutant loci on the *HOXC8* gene, and all of them were novel mutant loci. The types of thoracolumbar vertebrae in Dezhou donkeys were associated with carcass weight traits. The polymorphisms of the *HOXC8* gene were associated with its multi-vertebral number and carcass weight traits. The g.15179224C>T and g.15179674G>A polymorphisms significantly affected CW, LL, and LN traits. In addition, the *HOXC8* g.15179224C allele had a positive effect on CW and LL; the g.15179674G allele had a negative effect on LN. Therefore, g.15179224C>T was probably the restriction locus restricting the increment of CW and LL, and g.15179674G>A was a possible candidate gene for selection and breeding of Dezhou donkeys with multi-vertebra number. All these findings emphasized that the *HOXC8* gene was an essential candidate gene impacting multi-vertebral number and carcass weight traits, providing helpful information for the marker-assisted breeding of Dezhou donkeys.

## Figures and Tables

**Figure 1 genes-13-02175-f001:**
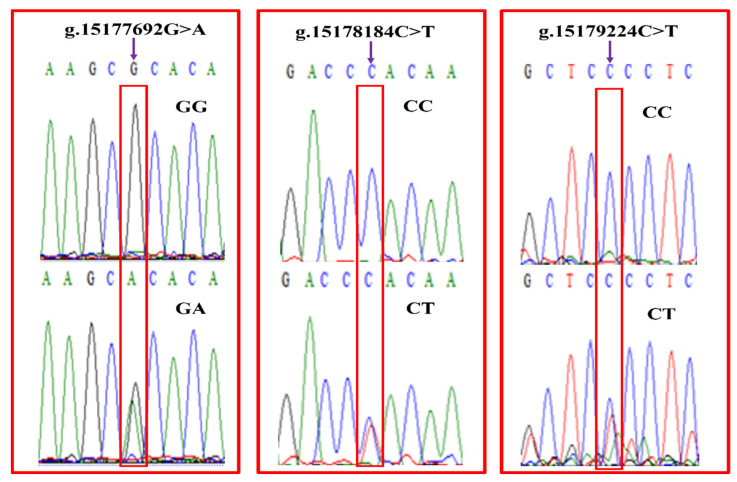
Sequencing profiles around three mutation loci in the *HOXC8* of Dezhou donkeys. The upper panels represent the reference genome, and the lower panels represent the variant.

**Figure 2 genes-13-02175-f002:**
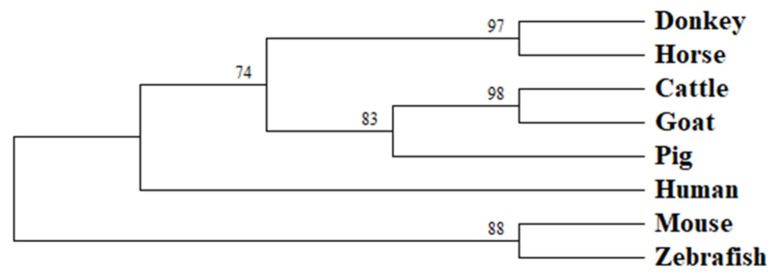
The phylogenetic tree of the *HOXC8* gene in different species. The number on the internal node represents support value and is used to represent the reliability of this branch structure.

**Table 1 genes-13-02175-t001:** Primer sequences used for single nucleotide polymorphisms (SNPs) validation.

SNP Position	Primer Sequence (5′-3′)	PCR Size (bp)	Ta (°C)
g.15177692G>A	F:5′-TTGGACCAGGAACAGAGCTG-3′	539	58
	R:5′-GTCGCTCAGAACTCACCATA-3′		
g.15178184C>T	F:5′-GACAGCAAAGGGGAGGAAGG-3’	317	58
	R:5′-TGCTAGGGTTAGTGTATGAGATTGA-3′		
g.15179224C>T	F:5′-CACTTCATCCTTCGGTTCTGGA-3′	507	60
	R: 5′-GCCACTCTGCACTTGTAAACA-3′		

Note: Ta, annealing temperature.

**Table 2 genes-13-02175-t002:** The information of 12 mutation loci in the *HOXC8* gene of Dezhou donkeys.

Region	Genomic Location	Wild-Type	Mutant	Mutation in Ensembl Database	Mutation in NCBI Database	Mutation in European Variation Archive	AA Coding Residue	Amino Acid Change
Downstream	15177467	G	A	NO	NO	NO	-	-
	15177692	G	A	NO	NO	NO	-	-
	15177975	T	TATTA	NO	NO	NO	-	-
	15178184	C	T	NO	NO	NO	-	-
	15178385	C	G	NO	NO	NO	-	-
	15178428	C	T	NO	NO	NO	-	-
	15178486	C	T	NO	NO	NO	-	-
	15178804	A	G	NO	NO	NO	-	-
	15178804	A	GG	NO	NO	NO	-	-
	15178806	G	C	NO	NO	NO	-	-
Intron	15179224	C	T	NO	NO	NO	-	-
	15179674	G	A	NO	NO	NO	-	-

**Table 3 genes-13-02175-t003:** Population genetic analysis of 4 SNPs of the *HOXC8* mutation in Dezhou donkeys.

Site	Location	Genotype Frequency			Allele Frequency		*PIC*	*Ne*	*Ho*	*H*	*He*	HWE	
												Chi-Square	*p* Value
g.15177692G>A	Downstream (dist = 1137)	GG (393)	GA (7)	AA (0)	G	A	0.018	1.018	0.983	0.050	0.017	0.031	0.860
		0.983	0.017	0.000	0.991	0.009							
g.15178184C>T	Downstream (dist = 645)	CC (373)	CT (27)	TT (0)	C	T	0.064	1.070	0.935	0.148	0.065	0.488	0.485
		0.933	0.067	0.000	0.966	0.034							
g.15179224C>T	Intron	CC (394)	CT (6)	TT (0)	C	T	0.014	1.015	0.985	0.044	0.015	0.023	0.880
		0.985	0.015	0.000	0.993	0.007							
g.15179674G>A	Intron	GG (373)	GA (25)	AA (2)	G	A	0.067	1.075	0.930	0.156	0.070	4.453	0.035
		0.932	0.063	0.005	0.964	0.036							

**Table 4 genes-13-02175-t004:** LSMs with standard errors of each phenotype for different genotypes of 4 SNPs in Dezhou donkeys.

Site	Genotype	BH/cm	BL/cm	CC/cm	HW/kg	CW/kg	LN	LL/cm	TN	TL/cm	TLN
g.15177692G>A	GG (393)	134.81 ± 0.26	132.55 ± 0.31	144.85 ± 0.26	24.15 ± 0.14	151.32 ± 0.97	5.21 ± 0.02	24.12 ± 0.11	17.86 ± 0.02	72.81 ± 0.18	23.07 ± 0.02
	GA (7)	135.71 ± 1.91	132.43 ± 2.32	146.43 ± 1.92	24.04 ± 1.07	155.29 ± 7.25	5.14 ± 0.15	23.57 ± 0.81	18.00 ± 0.14	73.14 ± 1.36	23.14 ± 0.13
g.15178184C>T	CC (373)	134.87 ± 0.26	132.60 ± 0.32	144.87 ± 0.26	24.12 ± 0.15	151.43 ± 0.99	5.21 ± 0.02	24.09 ± 0.11	17.86 ± 0.02	72.83 ± 0.19	23.07 ± 0.02
	CT (27)	134.22 ± 0.97	131.89 ± 1.18	144.98 ± 0.98	24.51 ± 0.54	150.83 ± 3.69	5.22 ± 0.08	24.30 ± 0.41	17.85 ± 0.07	72.63 ± 0.69	23.07 ± 0.07
g.15179224C>T	CC (394)	134.85 ± 0.25	132.59 ± 0.31	144.89 ± 0.26	24.17 ± 0.14	151.87 ± 0.95 ^A^	5.21 ± 0.02	24.14 ± 0.11 ^a^	17.86 ± 0.02	72.86 ± 0.18	23.07 ± 0.02
	CT (6)	133.33 ± 2.06	129.83 ± 2.50	144.00 ± 2.08	22.57 ± 1.15	119.92 ± 7.67 ^B^	5.17 ± 0.17	22.25 ± 0.87 ^b^	17.83 ± 0.16	70.33 ± 1.47	23.00 ± 0.14
g.15179674G>A	GG (373)	134.96 ± 0.26	132.72 ± 0.32	144.93 ± 0.26	24.15 ± 0.15	151.69 ± 0.99	5.21 ± 0.02 ^a^	24.10 ± 0.11	17.86 ± 0.02	72.81 ± 0.19	23.07 ± 0.02
	GA (25)	133.16 ± 1.01	130.08 ± 1.22	144.38 ± 1.02	23.91 ± 0.57	146.78 ± 3.84	5.20 ± 0.08 ^a^	23.79 ± 0.42	17.88 ± 0.08	73.03 ± 0.74	23.08 ± 0.07
	AA (2)	131.00 ± 3.56	131.00 ± 4.33	142.00 ± 3.60	22.15 ± 2.00	152.50 ± 13.56	6.00 ± 0.29 ^b^	27.50 ± 1.50	17.50 ± 0.27	72.50 ± 2.55	23.50 ± 0.24
overall average of phenotypic data		134.90 ± 0.25	132.59 ± 0.31	144.97 ± 0.26	24.18 ± 0.14	152.01 ± 0.89	5.21 ± 0.02	24.11 ± 0.11	17.86 ± 0.02	72.89 ± 0.18	23.07 ± 0.02

Note: Different letters for the groups indicate difference: (a, b) indicate differences (*p* < 0.05); (A, B) indicate differences (*p* < 0.01). BH, body height; BL, body length; CC, chest circumference; HW, hide weight; CW, carcass weight; LN, number of lumbar vertebrae; LL, lumbar vertebrae length; TN, number of thoracic vertebrae; TL, thoracic vertebrae length; TLN, total number of thoracic and lumbar vertebrae.

**Table 5 genes-13-02175-t005:** Means with standard error of each phenotype for different thoracolumbar types of Dezhou donkeys.

Type of Thoracolumbar	Number	BH/cm	BL/cm	CC/cm	HW/kg	CW/kg	Proportion
T17L5	10	131.90 ± 1.72	129.30 ± 1.41	141.10 ± 1.56	23.45 ± 0.55	127.20 ± 14.56 ^a^	2.50%
T17L6	50	134.64 ± 0.64	132.10 ± 0.72	144.48 ± 0.68	23.77 ± 0.38	150.09 ± 2.02 ^b^	12.50%
T18L5	301	134.90 ± 0.30	132.56 ± 0.36	145.09 ± 0.30	24.22 ± 0.17	152.25 ± 1.07 ^b^	75.25%
T18L6	34	135.38 ± 0.79	134.15 ± 0.98	144.81 ± 0.64	24.34 ± 0.47	152.71 ± 2.36 ^b^	8.50%
T19L5	5	134.60 ± 3.12	132.20 ± 3.85	144.40 ± 2.50	23.68 ± 1.17	151.50 ± 6.27 ^b^	1.25%

Note: Different letters for the groups indicate difference: (a, b) indicate differences (*p* < 0.05). BH, body height; BL, body length; CC, chest circumference; HW, hide weight; CW, carcass weight.

## Data Availability

The data presented in this study are available on request from the corresponding author.

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
