# Peer review of "Association of HOXC8 Genetic Polymorphisms with Multi-Vertebral Number and Carcass Weight in Dezhou Donkey"

_genes, 2022, doi:10.3390/genes13112175_

Round 1
Reviewer 1 Report
Line 16-19 Please paraphrase to show the importance of the research
In line 81 you reported “Blood samples of 400 adult male Dezhou donkeys” and in the statistical model you studied the effect of gender... Please revise…
Are there pedigree records for these animals?... If they exist, the sire effect should be included as a random effect
In line 130 ….you said and each observation's random error effect.. What do you mean by this sentence?
The other fixed factors should be incorporated in the statistical model such as birth weight, birth season and dam parity as classes, average daily gain as a covariate…. etc… not just animal age as you referred in order to adjust for environmental effects
You must insert a table to show the overall average of phenotypic data and then link that with the HOXC8 gene polymorphism.
Line 143-145. Please move these sentences into the M&M part
The numbers of samples must be included in Table 4.
With respected to the phylogenetic tree (Figure 2)... I think it is better to include it in a full research and not on the sidelines of a research topic like this
Reviewer 2 Report
This study presents a known gene studied in a new species, which make of interest for livestock production.
I found the document well done and conducted in terms of objectives and methodology. Conclusions were according to the presented results and discussion. The primary observation was in terms of format, especially in table 4. Authors can adjust the necessary number of decimals.Author Response
Please see the attachment.

Reviewer 3 Report
This article describes novel gene variants in HOXC8 in search for association with carcacss meat quality phenotypes in Dezhou Donkey. Over all the rational behind this study is justify and the methods used and the large sample size (n=400 samples) are good with few minor suggestions listed below. The authors found x12 gene variants in the sequenced samples and further explored x4 of these for association to meat phenotypes, including vertebra number.
There are some issues with interpretations of the results that I suggest to address:
1. The authors searched the identified gene variants against one database - Ensembl. There are several other gene databases of gene variants that it might be worthwhile to search and confirm novelity against these (e.g. European Variation Archive).
2 The authors identified x12 gene variants, but non of these had caused amino acids change in translation. Hence, the suggestions that two of these HOXc8 variants are significantly associated with increased vertebra number is not clear. It will be good to investigate if the same loci has other gene transcripts, or if this location also contains a miRNA, lnRNA or other regulatory elements that may explain such association (as the protein should not be changed by these variants). If no other explanation is identified, then the author suggestion might be speculative and should be analyzed differently.
3. It will aid to add a descriptive summary or table with the data on how much samples had the described phenotypes - specifically in regard to the LN phenotype - how much donkeys had larger number of vertebra and how many cervical/lumbar vertebra they had?
Below are also some minor comments:
4. List gene name in full when first introduce (e.g. Line 54 - NR6A1 & throughout the manuscript).
5. Line 87 - citation of methods from Zhang et al. - it will be helpful to summarise the key points in a sentence or two, to help the reader.
6. Line 132 - "the values in the Table..." - which table?
7. Lines 144-146 belong to the method section.
8. Figure 1 - It is not clear what is the upper and lower panels represent? (I assume one is the reference genome while the other is the variant?
9. Line 157 - Can the authors please clarify why the variants with extensive deletions were excluded?
10. Figure 2 - please explain in the legend what the values listed represent?
11. General comment - it will be worthwhile to specify if there is any risk or clinical signs that can be associated with a larger number of vertebra, or in other words is there a welfare issue in selecting for increased vertebrae length/number?
Round 2
Reviewer 1 Report
Line 140, the statistical model is not clear, it must contain numbering in alphabetical letters and an explanation of the letters used.I suggest adding the types of thoracolumbar as a fixed effect in the statistical model and delete the results in table 6 and referring to the proportions in the context in M&M part What is the statistical test used to detect the normal distribution of data? In lines 150 and 150 even though that you referred that the phenotypic data that did not meet the normal distribution were transformed using Box-Cox or Johnson to ensure that all data met the normal distribution, but in tables related to phenotypic data, there is no transformed data… please revise I do not see any importance of table number 5 in the search. You can delete it…… The table of overall average that I asked you to add before should include the overall average of the population not for each class that you added In the previous manuscript, you added the age in the statistical model to correct for its effects, but in the current manuscript you reported that all animals had the same age….. Please revise.Author Response
Please see the attachment.
